# Physicians' perception of task sharing with non-physician health care workers in the management of uncomplicated hypertension in Nigeria: A mixed method study

Oluseyi Ademola Adejumo[1]*, Olorunfemi Akinbode Ogundele[2], Manmak Mamven[3], Folajimi Morenikeji Otubogun[1], Oladimeji Adedeji Junaid[1], Ogochukwu Chinedum Okoye[4], Dapo Sunday Oyedepo[5], Osariemen Augustine Osunbor[6], Stanley Chidozie Ngoka[7], Adenike Christianah Enikuomehin[1], Kenechukwu Chukwuemeka Okonkwo[1], Akinwumi Ayodeji Akinbodewa[1], Olutoyin Morenike Lawal[1], Shamsuddeen Yusuf[8], Enajite Ibiene Okaka[9], Joseph Odu[10], Emmanuel Agogo[1], Kufor Osi[10], Ifeanyi Nwude[10], Augustine Nonso Odili[3]

1 Department of Internal Medicine, University of Medical Sciences, Ondo State, Nigeria, 2 Department of Community Medicine, University of Medical Sciences, Ondo State, Nigeria, 3 Department of Internal Medicine, University of Abuja, Gwagwalada, Nigeria, 4 Department of Internal Medicine, Delta State University, Oghara, Delta State, Nigeria, 5 Department of Internal Medicine, University of Ilorin, Kwara State, Nigeria, 6 Department of Internal Medicine, Stella Obasanjo Hospital, Benin City, Edo State, Nigeria, 7 Department of Internal Medicine, Federal University Teaching Hospital, Owerri, Imo State, Nigeria, 8 Department of Internal Medicine, Aminu Kano Teaching Hospital, Kano State, Nigeria, 9 Department of Internal Medicine, University of Benin, Edo State, Nigeria, 10 Resolve to Save Lives Organization, Nigeria

* oluseyiadejumo2017@gmail.com

## Abstract

### Introduction

Task sharing and task shifting (TSTS) in the management of hypertension is an important strategy to reduce the burden of hypertension in low-and middle-income countries like Nigeria where there is shortage of physicians below the World Health Organization's recommendations on doctor-patient ratio. The cooperation of physicians is critical to the success of this strategy. We assessed physicians' perception of TSTS with non-physician health workers in the management of hypertension and sought recommendations to facilitate the implementation of TSTS.

### Materials and methods

This was an explanatory sequential mixed method study. TSTS perception was assessed quantitatively using a 12-item questionnaire with each item assigned a score on a 5-point Likert scale. The maximum obtainable score was 60 points and those with ≥42 points were classified as having a good perception of TSTS. Twenty physicians were subsequently interviewed for in-depth exploration of their perception of TSTS.

### Results

A total of 1250 physicians participated in the quantitative aspect of the study. Among the participants, 56.6% had good perception of TSTS in the management of hypertension while

**Data Availability Statement:** All relevant data are within the paper and its Supporting Information files.

**Funding:** The authors received financial support from Nigerian Hypertension Society (PR0201). This supported data collection and analysis. The funders had no role in study design, data collection and analysis, decision to publish, or preparation of the manuscript.

**Competing interests:** The authors have declared that no competing interests exist.

about two-thirds (67.5%) agreed that TSTS program in the management of hypertension could be successfully implemented in Nigeria. Male gender (p = 0.019) and working in clinical settings (p = 0.039) were associated with good perception. Twenty physicians participated in the qualitative part of the study. Qualitative analysis showed that TSTS will improve overall care and outcomes of patients with hypertension, reduce physicians' workload, improve their productivity, but may encourage inter-professional rivalry. Wide consultation with stakeholders, adequate monitoring and evaluation will facilitate successful implementation of TSTS in Nigeria.

## Conclusion

This study showed that more than half of the physicians have good perception of TSTS in hypertension management while about two-thirds agreed that it could be successfully implemented in Nigeria. This study provides the needed evidence for increased advocacy for the implementation of TSTS in the management of hypertension in Nigeria. This will consequently result in improved patient care and outcomes and effective utilization of available health care personnel.

## Introduction

Hypertension is a disease of public health relevance with a high prevalence and associated huge burden of morbidity and mortality especially in low- and middle-income countries (LMIC) [1–5]. The World Health Organization's (WHO) report showed that hypertension affects about 1.3 billion adults worldwide [3]. A significant proportion of those with hypertension are unaware of their hypertensive state or not currently undergoing treatment [3, 6]. The burden of hypertension in Nigeria is also high with an overall age-standardized prevalence of 38.1% [6].

Hypertension and its related complications accounted for a significant number of hospitalizations, mortality and annual health cost over the years [7–12]. Adequate blood pressure control reduces the burden of hypertension [13]. However, reports showed that majority of patients with hypertension have suboptimal blood pressure control associated with complications that increase the health care expenditure [6, 14–17].

The burden of poor blood pressure control is higher in LMIC [15]. A recent report on hypertension control in 60 primary health centres in Nigeria showed that 89.2% of patients were on treatment. However, the baseline hypertension control rate was 13.1% [18]. There is currently a shortage of physicians in these countries, which is compounded by the emigration of health care workers to high-income countries [17, 19]. The need to reduce the burden of non-communicable diseases including hypertension, despite shortage of physicians in hospitals in these countries, thus becomes apparent and urgent. In the current context, there is need to adopt strategies that will facilitate early diagnosis and optimal treatment of hypertension especially in LMIC where health personnel shortage is more pronounced [20, 21].

Task sharing and shifting (TSTS) in management of uncomplicated hypertension is one of the effective strategies that may be employed in response to the shortage of health personnel. TSTS involves the redistribution or delegation of health care tasks within the workforce and communities in such a way that the quality of care is not affected. Task shifting occurs when a task is transferred or delegated while task sharing occurs when tasks are completed

collaboratively between providers with different levels of training [22]. The primary aim of this strategy is to effectively utilize existing human resource for health to be able to deliver quality health services to the population without compromising standards of care. In this case, the targeted non-physician health workers will undergo some specific trainings before they take up assigned tasks. These strategies have been widely used in tuberculosis programs, HIV care and maternal health services with positive outcomes in some countries including Nigeria [23–25]. Some previous systematic reviews and meta-analyses have reported that team-based approach in the management of hypertension achieved better blood pressure control compared to physician-centered care [26, 27]. Home based hypertension management by non-physicians have also been reported to be effective and efficient [28].

The TSTS for essential health care services policy in Nigeria that was approved in 2014 and updated in 2018, was only for communicable diseases prevention and control [29]. However, a recent policy has been formulated for non-communicable diseases, but is yet to be published. The TSTS strategy for non-communicable diseases especially for the management of essential hypertension is being piloted in Nigeria under the Hypertension Treatment in Nigeria (HTN) Program [30]. For decades, the management of hypertension rests solely on physicians and therefore, the cooperation of physicians as major stakeholders in this service delivery is crucial for the successful implementation of this strategy. It is against this background that this study aims to evaluate physician's views on the proposed strategy. This study determined the perception of physicians about TSTS with non-physician health workers in the management of uncomplicated hypertension. The findings of this study will be helpful in identifying areas that need to be addressed for a successful implementation of the strategy.

## Methods

This was an explanatory sequential mixed method study that used both quantitative and qualitative data collection to determine the perception of physicians towards TSTS with non-physician health workers in the management of uncomplicated hypertension in Nigeria. The study was conducted over an 8-week period between 27th July and 22nd September 2022.

### Sample size calculation

The sample size for the quantitative aspect of the study was determined using the formula for single proportion [31], and by assuming that the proportion of physicians with good perception of task sharing was 50%, an error margin of 5%, and a 95% confidence interval, giving a sample size of 385. A 10% non-response rate was anticipated. The total number of participants was therefore, 424. Only consenting, licensed and practising physicians who were either previously or currently involved in clinical work were included in the study.

### Sampling method

The quantitative arm of the study was an online survey that used purposive sampling method in recruiting physicians across all the geopolitical zones in Nigeria. The link to the online form was shared on social media platforms of various physician groups.

### Quantitative data collection

The questionnaire had two sections. The first section consisted of questions on socio-demographic information, cadre of respondent, area of specialization, hospital of practice, number of years post MBBS and fellowship qualification. The second section of the questionnaire assessed perception of task sharing for hypertension among respondents. Perceptions of TSTS

was assessed using 12 items that were combined into a single composite variable. Each item was assigned a score on a 5-point Likert scale (strongly disagree, 1; disagree, 2; neutral, 3; agree, 4; and strongly agree, 5) to capture the full range of opinions. Some of the items were reverse coded so that higher values indicated positive perceptions of task sharing for hypertension. A higher score indicated that the participant had positive perceptions of TSTS about uncomplicated hypertension. The maximum obtainable score was 60 points computed from the 12 items in the questionnaire. The mean score was 42 points, used as the cut-off for categorising the respondents [32]. Respondents who had score ≥42 were classified as having positive perception of TSTS in the management of uncomplicated hypertension while respondents who scored <42 were classified as having negative perception of TSTS in the management of uncomplicated hypertension. The questionnaire was pretested among 20 medical doctors who were not included in the study. Cronbach's α reliability coefficient was used to determine the reliability of the 12-item scale used as data collection instrument. It had high reliability, with a Cronbach's α of 0·85.

## Qualitative data collection

The interview questions and discussions were centred on the effect of current shortage of doctors on hypertension management in Nigeria; consequences of involvement of untrained non-physician health workers in hypertension management; consequences of TSTS in hypertension management on patients and physicians, medical profession and inter-professional rivalry; and recommendations to successfully implement TSTS in hypertension management in Nigeria.

The interviews were conducted by male and female consultant physicians after they had undergone training. A pilot study was done by interviewing 5 physicians which were not included in the main study. The result of the pilot study was reviewed by the researchers and some corrections were made to improve the clarity of the questions used for the interview. Key informant in-depth interviews were then conducted after the pilot study among twenty physicians selected using purposive sampling method between 17th and 22nd September, 2022. The participants were selected across the geopolitical zones of the country and involved different cadres of physicians.

The interview was face-to-face for 16 respondents while the remaining 4 had on-line interviews. The average duration of interview was about 15 minutes. The interviews were conducted in English language under conducive environment devoid of distractions following previously scheduled appointments. The interviews were audio recorded and transcribed. There were no repeat interviews and further interview was discontinued after saturation was reached. The authors ensured that the recorded interviews were correctly transcribed by listening to the recorded interview and matching with the transcription multiple times. The transcriptions were also reviewed by the interviewee to ensure there was no misrepresentation and feedbacks were obtained from them.

## Ethical consideration

Informed consent was obtained from all participants and their information was treated with utmost confidentiality. Ethical approval was obtained from National Health Research Ethics Committee of Nigeria, Federal Ministry of Health, Abuja, Nigeria. The approval number was NHREC/01/01/2007.

## Data analysis

Quantitative data was imported from an online excel sheet and analysed using IBM SPSS ver. 25.0 (IBM Corp., Armonk, NY, USA). Frequency and percentage distributions were used to

express socio-demographic variables. Perception of task sharing was presented as bar graph. The associations between socio-demographic variables and perception of task sharing were examined using the chi-square test. The outcome variable for the bivariate analyses was dichotomised into positive or negative perception. The independent variables (age, gender, specialty in medical field, years post qualification, etc.) were re-categorized into dichotomous variables for bivariate analyses. The level of significance for each test was set at $p < 0.05$.

Thematic analysis was used to analyze the interview data. This primarily involved evaluating the study transcripts for themes and patterns of meaning based on the research questions. This was done in six basic phases. First, the interview transcripts were read and reread in order to familiarize with the interview contents. Afterwards, the transcripts were coded based on emerging ideas from the interviews. The codes were later recoded in order to remove redundant ideas. The themes and subthemes were generated inductively. Subsequently, a report was written on the emergent themes. Three independent coders generated the initial set of themes and a fourth independent coder reviewed the themes and reorganized the thematic ideas. The analysis was done using ATLAS.ti version 22.2.4.

## Results

A total of 1, 250 physicians participated in the quantitative arm of the study with a mean age of 39±9.98 years. At the time of the study, 88% of the physicians were involved in clinical work. More than half of participants were below 39 years of age (50.9%) and were mostly male (64.2%). About one third of participants (37.9%) were less than 9 years post qualification while majority were between 10–19 years post qualification as indicated in Table 1. There were respondents from all the six geopolitical zones of Nigeria [Fig 1].

Overall, 56.6% of the participants had positive perceptions towards task sharing in the management of uncomplicated hypertension, while 43.4% had negative perceptions of task sharing with non-physicians. Specific analysis of the items within the scale revealed that, while 60.2%

**Table 1. Socio-demographic characteristics of study participants.**

| Variable | Frequency N = 1250 | Percent (%) |
|---|---|---|
| **Age (years)** | | |
| <39 | 636 | 50·9 |
| 40–59 | 555 | 44·4 |
| 60 and above | 59 | 4·7 |
| **Sex** | | |
| Male | 802 | 64·2 |
| Female | 448 | 35·8 |
| **Years post Qualification** | | |
| ≤9 | 473 | 37·9 |
| 10–19 | 516 | 41·2 |
| ≥20 | 261 | 20·9 |
| **Setting of Work** | | |
| Federal | 618 | 49·4 |
| State | 394 | 31·5 |
| Private | 236 | 1 8·9 |
| Mission | 2 | 0·16 |
| **Type of primary work** | | |
| Clinical work | 1100 | 88·0 |
| Others (admin, policy research, teaching) | 150 | 12·0 |

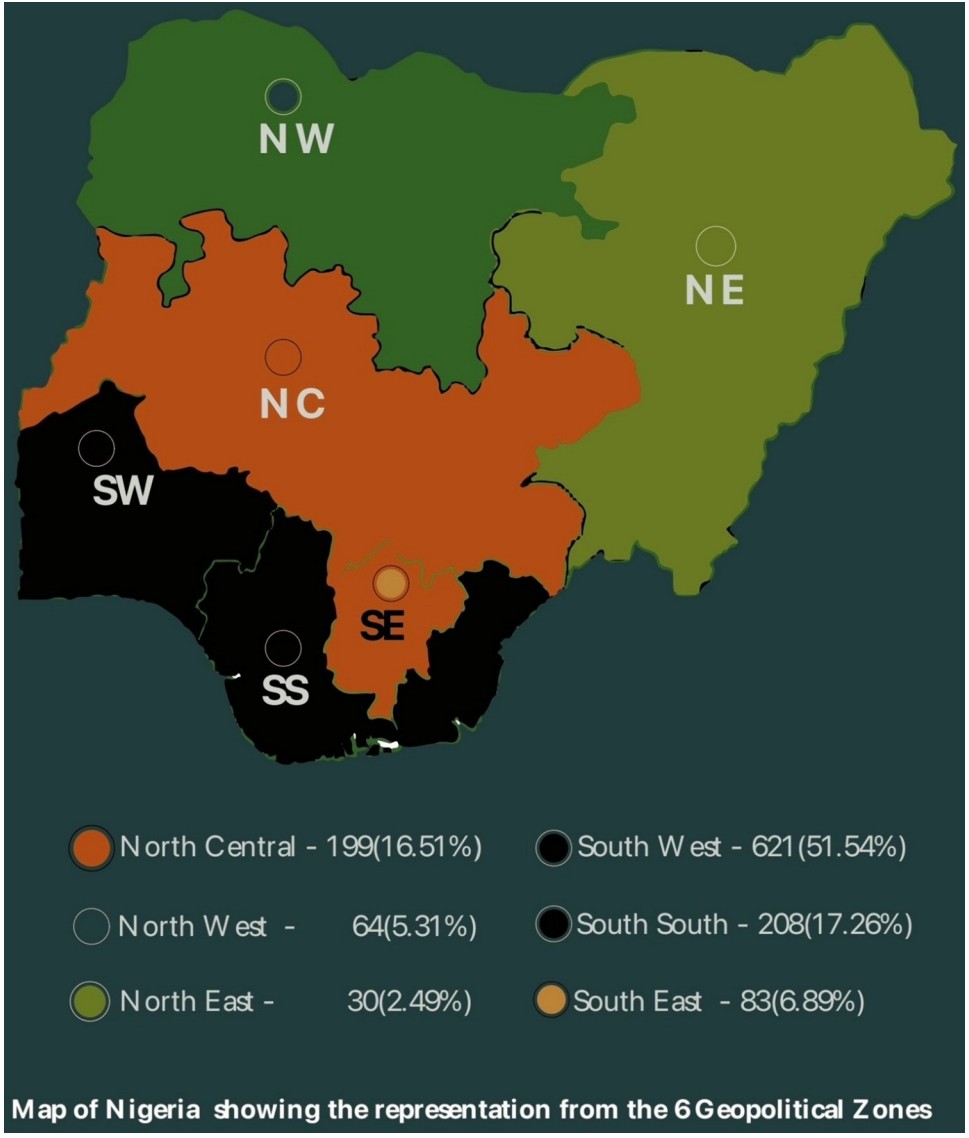

Fig 1. Map of Nigeria showing the representation from the six geopolitical zones.

agreed that due to the shortage of doctors, hypertension may not be optimally managed alone by doctors, about 89.7% of the physicians agreed and opined that involvement of non-physicians in the management of uncomplicated hypertension should be well monitored and supervised by Physicians [Fig 2].

Less than one-third of the respondents (26.5%) agreed that adequately trained non-physicians could be allowed to manage uncomplicated hypertension using protocols without supervision. More than half (58.5%) of the respondents agreed that involvement of non-physicians in management of uncomplicated hypertension will foster interdisciplinary cooperation and collaboration; will improve the quality of care of patients (53.6%); and will result in effective utilization of health personnel (64.5%). Less than half (42.4%) felt that it may encourage substandard medical practice in the country while about two-thirds (67.5%) agreed that TSTS program in the management of hypertension could be successfully implemented in Nigeria [Fig 2].

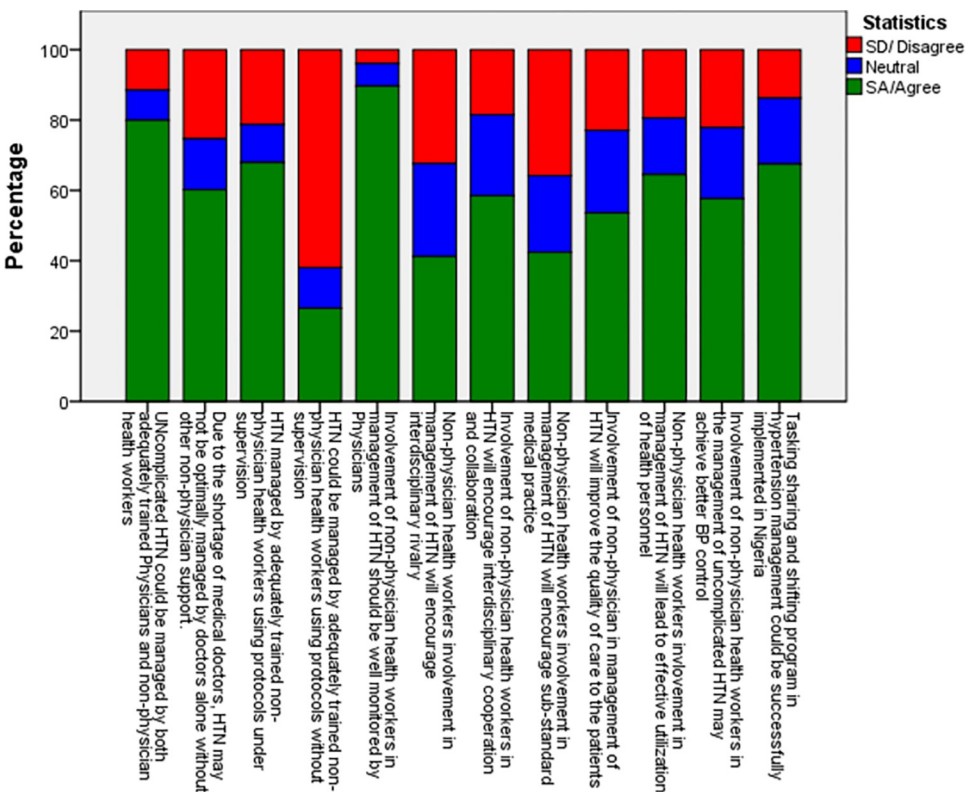

**Fig 2. Distribution of physician responses to the 12-item scale on perceptions of task sharing in uncomplicated hypertension management with non-physician.**

Bivariate assessment to determine factors associated with perception of TSTS in the management of uncomplicated hypertension among respondents revealed that a higher proportion of male physicians (59.1%) had positive perception of TSTS in the management of hypertension compared to the female physicians and the difference was statistically significant (p = 0.019). In addition, statistically significant association was also found between type of primary work and perception of task sharing (p = 0.036). A higher proportion of physicians (64.7%) who were engaged in clinical duties had a positive perception towards TSTS in the management of uncomplicated hypertension compared to those involved in non-clinical duties. Although a higher proportion of physicians in the private (61.9%), state (55.9%) settings and those who have practiced for less than 20 years (56.8%) had a positive perception towards task sharing in hypertension management, it was not statistically significant [Table 2].

## Qualitative analysis result

Twenty doctors made up of 11 males and 9 females participated in the key informant interviews. The theme and subtheme generated are presented in Table 3.

**Theme 1: Shortage of physicians is associated with negative health impacts.** The respondents perceived that the emigration of doctors and resultant shortage of physicians has led to poor access to physicians and reduction in quality of health services; poor clinic and medication compliance, poor blood pressure control, increased patronage of non-qualified personnel for care, and increase in hypertension-related morbidities and mortality.

**Table 2. Factors associated with perception of task sharing among participants.**

| Variable | Positive Perception n (%) | Negative Perception n (%) | P-value |
|---|---|---|---|
| **Age(years)** | | | |
| ≤39 | 370(58·2) | 226(41·8) | 0·265 |
| ≥40 | 338(55·0) | 276(45·0) | |
| **Sex** | | | |
| Male | 474(59·1) | 238(40·9) | 0·019 |
| Female | 234(52·2) | 214(47·8) | |
| **Years post Qualification** | | | |
| ≤19 | 562(56·8) | 427(43·2) | |
| ≥20 | 146(55·9) | 114(44·1) | 0·797 |
| **Settings of work** | | | |
| Federal | 334(54·0) | 284(46·0) | |
| State | 223(56·6) | 171(43·4) | 0·086 |
| Private | 146(61·9) | 90(38·1) | |
| Mission | 2(100·0) | 0(0·0) | |
| **Type of primary work** | | | |
| Clinical work | 612(55·6) | 488(44·4) | 0·036 |
| Others (admin, policy research, teaching etc.) | 97(64·7) | 53(35·3) | |

" *These few doctors available will not be able to attend to all the patients and some patients will seek for help elsewhere, by so doing, some of them may fall into the hands of quacks, people who are not doctors, who are not competent in the management of hypertension and it can have a disastrous effect on the patient.*" (Male, Consultant)

"*Many patients will end up being non-compliant with both clinic visits and possibly with medications because the mass movement of doctors out of the country has led to clinics becoming crowded, people will be wary of prolonged wait times.*" (Female, Senior Registrar)

**Theme 2: Involving untrained non-physician health workers in hypertension management will lead to poor health outcomes.** Some respondents opined that involving untrained non-physicians may lead to poor management, poor follow-up care, poor medication compliance and poor outcome of patients; increase in hypertension related complications, morbidity and mortality.

"*It only means most patients will not get the adequate care they require. They will get half-baked management since they are not trained and skilled in management of hypertension*" (Male, Principal Medical Officer)

"*Definitely the patient will suffer because, these non-physician's health workers do not have the experience and may increase complications and morbidity among patients.*" (Male, House Officer)

**Theme 3: Trained non-physician health workers will help improve outcomes of hypertension management.** Identified benefits to patients are better access to health care and information, better relationship and interactions with care provider, better blood pressure control and reduction in hypertension-related complications.

**Table 3. Themes and sub-themes.**

| Themes | Sub-themes |
|---|---|
| 1. Shortage of physicians is associated with negative health impacts | higher risk of death |
| | Increase in hypertension related complications |
| | increase in waiting time to physicians |
| | Encourages visits to quacks for medical care |
| | limited access to professional care |
| | Increase in patients' non-compliance |
| | Poor patients' management and outcomes |
| 2. Involving untrained non-physician health workers in hypertension management will lead to poor health outcomes | Poor patients' management and outcomes |
| | Increase in mortality |
| | Increase in hypertension related complications |
| | Poor compliance with treatment |
| | Poor follow-up of patients |
| 3. Trained non-physician health workers will help improve outcomes of hypertension management | Better access to health care |
| | Better access to health-related information |
| | Better relationship and interaction to health care provider |
| | Better blood pressure control |
| | Better compliance with medications |
| | Reduction in hypertension-related complications |
| 4. Trained non-physician health workers will reduce the work-related burden among physicians | Reduction of workload |
| | Reduction of work-related stress |
| | Reduction of work-related burnout |
| | Opportunity to dedicate time to more complex medical conditions |
| 5. Mixed effects on the medical practice | Improvement on overall health |
| | Improve inter-professional relationship |
| | Better health education of the public on hypertension |
| | Encourage unprofessional conduct |
| | Promote inter -professional rivalry |
| 6. Effective implementation involves planning and system approach | Wide consultation with stakeholders |
| | Clear job description |
| | Regular training |
| | Good referral system |
| | Regular monitoring and evaluation |
| | Working under supervision |
| | Development of simple treatment protocols and algorithms |
| | Appropriate sanctioning of erring workers |

*" It will improve access of patients to care and reduce the likelihood of complications. The non-physician health workers at the community know the patients' language and where they live. They can speak the language to them, they can interact with them. It's better for us, it's better"* (Male, Medical Director)

*".We are sure that they will get the right information. Some people don't like coming to the doctor for several reasons such as religious beliefs and what have you. Going to them too, you're getting the right information, you're getting the right drugs* "(Female, Medical Officer)

**Theme 4: Trained non-physician health workers will reduce the work-related burden among physicians.** The highlighted benefits include reduction of workload and related stress burnout, opportunity to dedicate time to see more challenging medical cases and having time to play other non-clinical roles.

*"Yes, I think they will benefit in the sense that their work will be focused on those that need more attention, like those that already have end-organ damage, and they will have less work load in the clinic"* (Female, Senior Registrar)

*"Burnout will be reduced because you have passed the bulk to other people, so you'll be able to concentrate on things that really matter."* (Male, Consultant)

**Theme 5: Mixed effects on the medical practice.** Some respondents believed that task sharing will have positive impact on the medical profession such as improvement on patients' overall health and inter-professional relationship, while others believed that it could encourage quackery and may embolden the non-physicians to go beyond the level of their training and competence to manage complicated medical conditions.

*"It will improve inter-professional relationships because sometimes people tend to feel doctors try to be lords over them. So having something like this will also improve interdependence among health workers, its good, it will improve working relationship between doctors and other health workers definitely"* (Male, Medical Officer)

*"As it is now we have a lot of people who are practicing as quacks and this may encourage more of that, and that's why I said that it will require a lot of supervision and there will have to be a major control, check and balances so that people don't just come up claiming that they have been trained."* (Female, Senior Registrar)

**Theme 6: Effective implementation involves planning and systemic approach.** The suggested recommendations for successful implementation of task sharing include wide consultation with all stakeholders such as health professionals and their associations, the general public; regular training; proper job description; regular supervision; monitoring and evaluation of the policy; good referral system; use of treatment protocols and algorithms; and appropriate sanctions to erring health workers.

*"If you are in a team of management of uncomplicated hypertension, let everybody know their job description. A well outlined job description will help and they know that if you step out of your boundary, you do things that you are not supposed to do, you are liable to punishment."* (Male, Medical Officer)

*"We should train them and let them know their limitations. They can use algorithm in patient care that tells them which group of patients they can manage and not manage; when to refer; who to refer to or the superior to call next. This has been used with positive results in obstetrics care in Ondo State*

We can now have protocols." (Male, Medical Director)

*"What you are doing now is number one. Get the opinions of the health workers; not only the doctors but other health workers such as pharmacists, nurses and lab scientists. We should also seek the opinions of the unions because in Nigeria, you can't run away from unions like the Nigeria Medical Association, Medical and Dental Consultants of Nigeria, the Nurses Union. Let them be well represented"* (Male, Director of Health Services)

## Discussion

This study assessed physicians' perception of TSTS with non-physician health workers in the management of hypertension and sought their recommendations for successful implementation of this policy in Nigeria. More than half of the physicians had a good perception of task sharing and about two-thirds believed that it could be implemented successfully in Nigeria.

Most of the respondents agreed that with the current emigration of physicians out of Nigeria, there is an increasing shortage of physicians with consequent adverse effects on the quality of health services in Nigeria. Reports showed that the present doctor-patient ratio in Nigeria is 1:5,000 as against 1:600 recommended by the WHO [33]. This implies that quality health care service delivery may not be possible for the majority of the Nigerian population.

Majority of physicians agreed that the burden of hypertension is high and a significant proportion of those with hypertension are currently being managed by non-health workers and non-physician health workers who have not received requisite training on hypertension management. They also agreed that this trend may have a negative impact on the outcomes of such patients such as suboptimal blood pressure control, poor adherence, increase in hypertension related complications and mortality. This corroborates a study done in Bangladesh where 40.7% of those with hypertension were diagnosed and managed by untrained non-healthcare personnel and these patients were more likely to be non-adherent with blood pressure medications possibly contributing to poor outcomes [34].

Most of the physicians agreed that many patients with hypertension in Nigeria have suboptimal blood pressure control. This is similar to previous reports within and outside Nigeria [6, 14, 15, 18]. Factors that have been reported to be associated with suboptimal control include physicians' inertia to review medications, poor medication adherence, wrong beliefs about hypertension management, lack of motivation, presence of co-morbidities, poor patient education and inaccessibility to specialist physicians [35, 36]. Adequate blood pressure control is pivotal to reducing hypertension associated morbidity and mortality which is still high especially in developing countries like Nigeria where there are limited facilities and low financial resources to adequately manage complications. Previous studies have reported that involvement of non-physician health workers in hypertension management led to better blood pressure control [26–28].

About two-thirds of the physicians agreed that implementation of TSTS with non-physician health workers will be beneficial to the patients, physicians and the entire health care system in Nigeria. It was also observed that a higher proportion of physicians involved in clinical activities had a good perception of TSTS compared to physicians involved in non-clinical activities. This may be due to the fact that those involved in clinical activities were more likely to appreciate the relevance of TSTS in patients' outcomes and physicians' productivity. Majority of participants in this study opined that implementation of TSTS may likely reduce workload of available physicians, give them the opportunity to pay adequate attention to more complicated and challenging medical conditions, reduce the likelihood of physician burnout and allow for efficient utilization of available health care personnel in the face of the current shortage of health care workers in the country. Onyiego et al [37]. reported that implementation of TSTS

with community health volunteers in Kenya in the management of hypertension significantly addressed health care workers' shortage with no adverse events. Some of the respondents noted that non-physician health workers, especially the community health workers have better and closer relationships with the patients, live with patients within the community, are in a better position to follow up the patients and regularly educate them on lifestyle modification and compliance with their medications through home visits. Ruilope et al. [38] specifically identified poor patient-physician relationship as an important factor associated with poor adherence to blood pressure medications among patients with hypertension.

About 43% of respondents had negative perception of TSTS and some negative opinions were expressed during the key informant interview. Some respondents were of the opinion that TSTS will promote interprofessional rivalry amongst the health workers; some non-physicians may practice beyond their level of training and capacity; and unhealthy competition between non-physician health workers and physicians may be encouraged.

Inter professional rivalry and associated problems are important causes of conflict within the health sector in Africa according to previous studies [39–41]. This reduces the overall productivity of the health team and prevents the patients from getting optimal care. Some respondents were however, of the opinion that inter professional collaboration in the health sector in Nigeria can be fostered if TSTS is properly implemented. Overall, our finding suggests that TSTS may be embraced and supported by more physicians if their concerns could be addressed.

Majority of the physicians in this study agreed that TSTS could be successfully implemented in Nigeria. Development of clear and detailed policy document and management protocols, clear job description, training, supportive supervision, regular monitoring and evaluation of the policy and involvement of other stakeholders in health care system such as professional associations of both physician and non-physician health workers were the common recommendations made by the physicians for successful implementation of TSTS in the management of hypertension in Nigeria. Some of these recommendations are similar to findings of a previous study that emphasized training, supervision and quality assurance as factors that will contribute to successful implementation of task sharing in hypertension management as in those with HIV [42].

Wide consultation with stakeholders such as the various categories of health professionals, health professional bodies and the general public cannot be overemphasized. This is a major lesson from the failed attempt by the Kenyan Ministry of Health to introduce TSTS policy in certain aspects of the country's health care services delivery. The implementation of a painstakingly developed task sharing policy was put on hold by the court after a legal tussle which ensued between Kenyan Ministry of Health and a health professional due to non-involvement of his professional association and the general public in the process of formulating the policy [43, 44]. The court not only declared the policy unconstitutional, but went ahead to grant a permanent injunction to restrain the Ministry from implementing the policy in question.

Previous studies have also harped on the need to adequately train non-physician health workers as it has been established that the quality and level of training and education of non-physicians is a determinant of outcome of TSTS in hypertension management [26, 27]. The HTN Program, a pilot study involving non-physician health workers in the management of hypertension in Nigeria is ongoing [30]. The preliminary findings from the HTN Program provides evidence on the pivotal role of non-physician health workers and team-based approach in the successful management of hypertension.

The limitation of this study was that comprehensive assessment of knowledge of TSTS and its association with their perception were not assessed among the respondents. The strength of this study lies in its mixed method approach which affords the opportunity for wide and in-

depth exploration of the research questions, thus creating reliable, credible, objective and transferable information. This study also involved a large sample size with respondents from all the six geopolitical zones of Nigeria, thereby making the findings of the study easily generalizable to the country and other similar contexts.

In conclusion, this study showed that more than half of the physicians have a good perception of TSTS in hypertension management while about two-thirds agreed that it could be successfully implemented in Nigeria. This study provides the needed evidence for increased advocacy for the implementation of TSTS in the management of hypertension in Nigeria. This will consequently result in improved patient care and outcomes and effective utilization of available health care personnel. We recommend that TSTS should be adopted in the management of non-communicable diseases such as hypertension especially in low resource countries like Nigeria. All critical stakeholders such as physicians, non-physician health workers and the general public should be involved in the policy formulation and implementation process.

## Supporting information

**S1 Checklist. COREQ checklist.**
(PDF)

**S1 File. Qualitative study report.**
(ZIP)

## Acknowledgments

The authors acknowledge the efforts of physicians who assisted in data collation.

## Author Contributions

**Conceptualization:** Oluseyi Ademola Adejumo, Olorunfemi Akinbode Ogundele, Manmak Mamven, Folajimi Morenikeji Otubogun, Oladimeji Adedeji Junaid, Ogochukwu Chinedum Okoye, Dapo Sunday Oyedepo, Osariemen Augustine Osunbor, Stanley Chidozie Ngoka, Adenike Christianah Enikuomehin, Kenechukwu Chukwuemeka Okonkwo, Akinwumi Ayodeji Akinbodewa, Enajite Ibiene Okaka, Augustine Nonso Odili.

**Data curation:** Oluseyi Ademola Adejumo, Olorunfemi Akinbode Ogundele, Manmak Mamven, Folajimi Morenikeji Otubogun, Ogochukwu Chinedum Okoye, Dapo Sunday Oyedepo, Osariemen Augustine Osunbor, Stanley Chidozie Ngoka, Adenike Christianah Enikuomehin, Kenechukwu Chukwuemeka Okonkwo, Olutoyin Morenike Lawal, Shamsuddeen Yusuf, Enajite Ibiene Okaka.

**Formal analysis:** Oluseyi Ademola Adejumo, Olorunfemi Akinbode Ogundele, Osariemen Augustine Osunbor.

**Funding acquisition:** Manmak Mamven, Joseph Odu, Emmanuel Agogo, Augustine Nonso Odili.

**Methodology:** Oluseyi Ademola Adejumo, Folajimi Morenikeji Otubogun, Oladimeji Adedeji Junaid, Ogochukwu Chinedum Okoye, Dapo Sunday Oyedepo, Osariemen Augustine Osunbor, Stanley Chidozie Ngoka, Adenike Christianah Enikuomehin, Kenechukwu Chukwuemeka Okonkwo, Akinwumi Ayodeji Akinbodewa, Olutoyin Morenike Lawal, Enajite Ibiene Okaka, Kufor Osi.

**Project administration:** Oluseyi Ademola Adejumo, Joseph Odu, Ifeanyi Nwude.

**Resources:** Oladimeji Adedeji Junaid, Adenike Christianah Enikuomehin, Joseph Odu, Ifeanyi Nwude, Augustine Nonso Odili.

**Supervision:** Oluseyi Ademola Adejumo, Oladimeji Adedeji Junaid, Dapo Sunday Oyedepo, Kenechukwu Chukwuemeka Okonkwo, Shamsuddeen Yusuf, Emmanuel Agogo, Kufor Osi, Ifeanyi Nwude, Augustine Nonso Odili.

**Writing – original draft:** Oluseyi Ademola Adejumo, Adenike Christianah Enikuomehin.

**Writing – review & editing:** Oluseyi Ademola Adejumo, Olorunfemi Akinbode Ogundele, Manmak Mamven, Folajimi Morenikeji Otubogun, Oladimeji Adedeji Junaid, Ogochukwu Chinedum Okoye, Dapo Sunday Oyedepo, Osariemen Augustine Osunbor, Stanley Chidozie Ngoka, Kenechukwu Chukwuemeka Okonkwo, Akinwumi Ayodeji Akinbodewa, Olutoyin Morenike Lawal, Shamsuddeen Yusuf, Enajite Ibiene Okaka, Joseph Odu, Emmanuel Agogo, Kufor Osi, Ifeanyi Nwude, Augustine Nonso Odili.

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
