## [Decision Letter · Decision Letter 0]

24 Apr 2023

PONE-D-22-34586Physicians’ Perception of Task Sharing with Non-Physician Health Care Workers in the Management of Uncomplicated Hypertension in Nigeria: a mixed method studyPLOS ONE

Dear Dr. ADEJUMO,

Thank you for submitting your manuscript to PLOS ONE. After careful consideration, we feel that it has merit but does not fully meet PLOS ONE’s publication criteria as it currently stands. Therefore, we invite you to submit a revised version of the manuscript that addresses the points raised during the review process.

Before our final decision, please resubmit your manuscript  with point-by-point responses to the editor and reviewers(your MS needs major revision, follow journal requirements and our publication criteria).

The scale utilized in the study needs to be revised as per the reviewer’s suggestion.Make corrections/use the standard language, refrain from typos/brevity, vague terminologies . Also engage language editor/sRewrite the results section as suggestedRevise the discussion section based on your  findings, closer to data and refrain from personal/researcher biasRevise conclusions and recommendations, based on evidence and anchored in scientific inquiryAmend references as suggestedPlease submit your revised manuscript by Jun 08 2023 11:59PM. If you will need more time than this to complete your revisions, please reply to this message or contact the journal office at plosone@plos.org. Please include the following items when submitting your revised manuscript:A rebuttal letter that responds to each point raised by the academic editor and reviewer(s). You should upload this letter as a separate file labeled 'Response to Reviewers'.A marked-up copy of your manuscript that highlights changes made to the original version. You should upload this as a separate file labeled 'Revised Manuscript with Track Changes'.An unmarked version of your revised paper without tracked changes. You should upload this as a separate file labeled 'Manuscript'.

We look forward to receiving your revised manuscript.

Kind regards,

Philipos Petros Gile, MA

Academic Editor

PLOS ONE

Journal Requirements:

3. We note that Figure 1 in your submission contain map images which may be copyrighted. All PLOS content is published under the Creative Commons Attribution License (CC BY 4.0), which means that the manuscript, images, and Supporting Information files will be freely available online, and any third party is permitted to access, download, copy, distribute, and use these materials in any way, even commercially, with proper attribution. For these reasons, we cannot publish previously copyrighted maps or satellite images created using proprietary data, such as Google software (Google Maps, Street View, and Earth). For more information, see our copyright guidelines: http://journals.plos.org/plosone/s/licenses-and-copyright.

(1) You may seek permission from the original copyright holder of Figure 1 to publish the content specifically under the CC BY 4.0 license.  

Reviewers' comments:

Reviewer's Responses to Questions

**Comments to the Author**

1. Is the manuscript technically sound, and do the data support the conclusions?

Reviewer #1: Yes

Reviewer #2: Yes

2. Has the statistical analysis been performed appropriately and rigorously? 

Reviewer #1: Yes

Reviewer #2: Yes

3. Have the authors made all data underlying the findings in their manuscript fully available?

Reviewer #1: Yes

Reviewer #2: Yes

4. Is the manuscript presented in an intelligible fashion and written in standard English?

Reviewer #1: Yes

Reviewer #2: Yes

5. Review Comments to the Author

Reviewer #1: My only sugestion is to check some typos in the tables and in some parts of the text. It seems that some symbols were missing. In general, the paper deals with an important issue in less developed countries and in most countries. The use of trained health workers, even not physicians may help to reduce the burden.

Reviewer #2: Good job by the authors. Made a good read on a germane issue and I appreciate the time and effort you have put into conducting your research and presenting your findings.

1. However, I noticed quite a few grammatical and typographical errors that may detract from the overall quality of the work. These affect the clarity and coherence of your arguments and distract readers from the substance of the research. I believe these should be corrected.

2. In addition, usually Task shifting and Task Sharing acronym is TSTS or TS/S. Why did the authors use TSSH?

3. The team mentioned 1250 doctors interviewed but the sample size mentioned in the methods section was much lower, please explain.

4. The Likert scale utilized in the study identified a score of 42 and above as indicative of a positive perception, as it was the mean value of the dataset. However, it is suggested that the scale could have been partitioned into three categories rather than two, with positive perception, negative perception and neutral perception based on the distribution of scores for more detailed analysis rather than classifying good and bad just by a point difference between them on a scale.

5. The Results sections should be rewritten as it is quite verbose with long sentences by the respondents which do not add to the analysis and could be summarized under the themes with some more analysis in the section. The tables could also do with some re-arrangement for ease.

6. Under the discussion section, please explain what you mean by brain drain and quacks to assist with clarity, in addition to some other terms in the text for an international audience. This is especially important to remove any suggestions of inherent bias since majority of the authors are doctors. In addition, distinguishing health trained versus non health trained staff, and which health workers can carry out management of hypertension in low resource settings such as where the study took place compared to tertiary health care settings could do with a literature check.

7. Line 341-The authors say *Task sharing is therefore, likely to address the consequences of poor patient-physician relationship*. This statement will benefit from a reference, or it should be reviewed.

8. The Conclusions and recommendations (381-386) need to be reviewed completely as recommendations based on interviewees perceptions are quite different from author recommendations based on scientific enquiry and findings as well as the literature review. The section requires a complete rewrite and needs to be organised.

9. There are a number of references of work done on hypertension in Nigeria recently that should be reviewed and considered to improve the document, especially with relation to lower cadre health workers in the area of hypertension.

10. The Nigerian TSTS guidelines have been reviewed and the changes should be included in the paper with relation to various cadres of health workers in the literature review section so that the article is not outdated even prior to publishing.

With improvements, I believe that your paper has the potential to make a significant contribution to the field of study around health policy formulation, training of health workers at various levels and the review of the TSTS policy in resource constrained settings.

6. PLOS authors have the option to publish the peer review history of their article (what does this mean?). If published, this will include your full peer review and any attached files.

Reviewer #1: No

Reviewer #2: No

---

## [Author Response · Author response to Decision Letter 0]

5 May 2023

We appreciate the comments and suggestions received from the reviewers towards improving the quality of our manuscript.

 REVIEWERS’ COMMENTS AUTHORS’ COMMENTS

1 My only suggestion is to check some typos in the tables and in some parts of the text. It seems that some symbols were missing This has been done

2 However, I noticed quite a few grammatical and typographical errors that may detract from the overall quality of the work. These affect the clarity and coherence of your arguments and distract readers from the substance of the research. I believe these should be corrected.

 This has been done

3 In addition, usually Task shifting and Task Sharing acronym is TSTS or TS/S. Whydid the authors use TSSH? This has been changed to TSTS

4 The team mentioned 1250 doctors interviewed but the sample size mentioned in the methods section was much lower, please explain. The study was a mixed method study. 1250 doctors participated in the online survey(quantitative part) while 20 doctors were interviewed for the qualitative part of the study

5 The Likert scale utilized in the study identified a score of 42 and above as indicative of a positive perception, as it was the mean value of the dataset. However, it is suggested that the scale could have been partitioned into three categories rather than two, with positive perception, negative perception and neutral perception based on the distribution of scores for more detailed analysis rather than classifying good and bad just by a point difference between them on a scale Thank you once again for this suggestion. We opted to use the positive and negative perception for a concise discussion while we acknowledge the argument about using midpoints opinions on the Likert scale and the epistemological connotation of such. 

However, the analysis decision was based on suggestions from previous literature on analyzing Likert items and the Likert scale amidst many arguments and what we felt was best in the context of the objective of the study. , , , 

We combined the items in order to generate a composite score (Likert scale) of the 12 items for the different participants, this thus enabled us to turn the assigned scale into an interval scale which we partitioned into two because we felt it was more beneficial to report the perception of the physician as positive or negative as that has significant implication for care and programme planning considering the importance of the topic rather than reporting as an ordinal variable with the midpoint opinions. 

We agree that the scale could be partitioned in the manner suggested; however, as said earlier we decided based on the abovementioned facts. In addition, we ensured we created Likert scale items by calculating a composite score (sum and mean) from the 12 Likert items so we can use parametric statistics such as mean for central tendency and standard deviation for variance in keeping with research based on Likert items and scales in previous literature that treats them as interval scales and analyzes them as such with descriptive statistics like means, standard deviations, etc. and inferential statistics.

Sullivan GM, Artino AR Jr. Analyzing and Interpreting Data from Likert-Type Scales. J of Grad Med Educ 2013DOI: http://dx.doi.org/10.4300/JGME-5-4-18

2 Subedi BP.Using Likert Type Data in Social Science Research: Confusion, Issues and Challenges. Inter J Con App Sci. 2016:3(2). www.ijcas.net

3 Rickards G, Magee C, Artino AR Jr. You can’t fix by analysis what you’ve spoiled by design: developing survey instruments and collecting validity evidence. J Grad Med Educ. 2012;4(4):407–41

4 Draeger, R. Cut-off scores. In M. Allen (Ed.), The sage encyclopaedia of communication research methods 2017: pp. 326-328. SAGE Publications, Inc, https://dx.doi.org/10.4135/9781483381411.n122

6 The Results sections should be rewritten as it is quite verbose with long sentences by the respondents which do not add to the analysis and could be summarized under the themes with some more analysis in the section. The tables could also do with some re-arrangement for ease. This has been revised as suggested

7 Under the discussion section, please explain what you mean by brain drain and quacks to assist with clarity, in addition to some other terms in the text for an international audience. This is especially important to remove any suggestions of inherent bias since majority of the authors are doctors. In addition, distinguishing health trained versus non health trained staff, and which health workers can carry out management of hypertension in low resource settings such as where the study took place compared to tertiary health care settings could do with a literature check

 Brain drain has been replaced

Quack has been replaced

The emphasis of our study was non-physician health workers

8 Line 341-The authors say *Task sharing is therefore, likely to address the consequences of poor patient-physician relationship*. This statement will benefit from a reference, or it should be reviewed This has been reviewed. 

The statement has been expunged

9 The Conclusions and recommendations (381-386) need to be reviewed completely as recommendations based on interviewees perceptions are quite different from author recommendations based on scientific enquiry and findings as well as the literature review. The section requires a complete rewrite and needs to be organised.

 The conclusion and recommendation have been revised as suggested

10 There are a number of references of work done on hypertension in Nigeria recently that should be reviewed and considered to improve the document, especially with relation to lower cadre health workers in the area of hypertension.

 More literature from Nigeria has been added

11 The Nigerian TSTS guidelines have been reviewed and the changes should be included in the paper with relation to various cadres of health workers in the literature review section so that the article is not outdated even prior to publishing.

 Thank you for this observation and suggestion.

We have provided more information on TSTS policy on communicable

However, on further enquiry we found that the TSTS policy document on non-communicable disease is yet to be published officially. It is being considered administratively at the Federal Ministry of Health

---

## [Decision Letter · Decision Letter 1]

17 Jul 2023

PONE-D-22-34586R1Physicians’ Perception of Task Sharing with Non-Physician Health Care Workers in the Management of Uncomplicated Hypertension in Nigeria: a mixed method studyPLOS ONE

Dear Dr. Adejumo, 

Thank you for submitting your manuscript to PLOS ONE. After careful consideration, we feel that it has merit but does not fully meet PLOS ONE’s publication criteria as it currently stands. Therefore, we invite you to submit a revised version of the manuscript that addresses the points raised during the review process. Please submit your revised manuscript by Aug 31 2023 11:59PM.  If you will need more time than this to complete your revisions, please reply to this message or contact the journal office at plosone@plos.org. Please include the following items when submitting your revised manuscript:A rebuttal letter that responds to each point raised by the academic editor and reviewer(s). You should upload this letter as a separate file labeled 'Response to Reviewers'.A marked-up copy of your manuscript that highlights changes made to the original version. You should upload this as a separate file labeled 'Revised Manuscript with Track Changes'.An unmarked version of your revised paper without tracked changes. You should upload this as a separate file labeled 'Manuscript'.

We look forward to receiving your revised manuscript.

Kind regards,

Philipos Petros Gile, MA

Academic Editor

PLOS ONE

**Additional Editor Comments:**

The manuscript needs major revision and point-by-point response to the reviewers with tidier changes as per the comments and improvement suggestions as a requirement for acceptance.

Reviewers' comments:

Reviewer's Responses to Questions

**Comments to the Author**

1. If the authors have adequately addressed your comments raised in a previous round of review and you feel that this manuscript is now acceptable for publication, you may indicate that here to bypass the “Comments to the Author” section, enter your conflict of interest statement in the “Confidential to Editor” section, and submit your "Accept" recommendation.

Reviewer #1: All comments have been addressed

Reviewer #2: All comments have been addressed

Reviewer #3: (No Response)

2. Is the manuscript technically sound, and do the data support the conclusions?

Reviewer #1: Yes

Reviewer #2: (No Response)

Reviewer #3: No

3. Has the statistical analysis been performed appropriately and rigorously? 

Reviewer #1: N/A

Reviewer #2: (No Response)

Reviewer #3: Yes

4. Have the authors made all data underlying the findings in their manuscript fully available?

Reviewer #1: Yes

Reviewer #2: (No Response)

Reviewer #3: Yes

5. Is the manuscript presented in an intelligible fashion and written in standard English?

Reviewer #1: Yes

Reviewer #2: (No Response)

Reviewer #3: Yes

6. Review Comments to the Author

Reviewer #1: The authors have made important improvements and I still consider that the text is an interesting contribution

Reviewer #2: (No Response)

Reviewer #3: Introduction

1. Line 83-84: The World Health Organization’s (WHO) report showed that hypertension affects about 1.3 billion adults worldwide – this statement needs to be cited.

2. Line 95: The word ‘brain drain’ still appears (despite previous assertions that it was replaced). Is this intentional?

Methods

1. Satisfactory

Results

1. Line 197: Clarify whether the 1250 were for just the quantitative aspect of the study or for both qualitative and quantitative.

2. Line 202 -203: A significant proportion (43.4%) have a negative perception about TSTS. This will suggest that the support is not overwhelming. Their concerns need to be explored to identify what points needs to be improved on.

3. Line 204 -208: This needs to be clarified; the percentages of those who believe trained non-physician HCWs need or do not need supervision is at variance/ confusing.

4. Line 208 -212: It will be beneficial to also state the percentages here.

Also, was the level of comprehension/ understanding of the TSTS concept among the respondents assessed? It will have been useful to compare their level of understanding of the concept with their perception. Because, if they had a good understanding of the concepts, and still did not give an enthusiastic endorsement, then there are clear concerns.

5. In addition, it will have been beneficial to narrow the respondents of this study to those who are involved, in some capacity, in the treatment of patients with hypertension.

6. Line 213 -222: These parameters used for the analysis here are rather generic for the evaluation of such an important policy.

i. What proportion of physicians recruited in this study are involved in clinical work?

ii. It would have been more informative if the inclusion criteria for this study had been better defined at the beginning.

Discussion

1. The discussion highlights the findings of positive perception among the respondents, without addressing the reasons for the significant proportion of respondents with negative impression.

In order to get the best out of this policy, the concerns of those with negative perception will need to be highlighted and addressed.

2. Also, it is worthy to note that the loss of HCWs in the country is not just among physicians, but other professionals cadres as well. What cadre do the authors propose in this roll-out?

7. PLOS authors have the option to publish the peer review history of their article (what does this mean?). If published, this will include your full peer review and any attached files.

Reviewer #1: **Yes: **Andres Navarro

Reviewer #2: No

Reviewer #3: No

---

## [Author Response · Author response to Decision Letter 1]

29 Jul 2023

REVIEWERS’ COMMENTS AUTHORS’ RESPONSE

1 Line 83-84: The World Health Organization’s (WHO) report showed that hypertension affects about

1.3 billion adults worldwide – this statement needs to be cited.

 Reference added

2 Line 95: The word ‘brain drain’ still appears (despite previous assertions that it was replaced). Is

this intentional? Brain drain has been deleted

3 Line 197: Clarify whether the 1250 were for just the quantitative aspect of the study or for both

qualitative and quantitative.

. 

 1250 was for quantitative. This has been clarified in the method

4 Line 202 -203: A significant proportion (43.4%) have a negative perception about TSTS. This will

suggest that the support is not overwhelming. Their concerns need to be explored to identify what

points needs to be improved on In the study, we explored various concerns, including the one mentioned above, through the qualitative component of the study. It's important to note that perception can vary depending on the context, which is why we conducted the qualitative assessment. The qualitative component sheds light on this.The Theme number 5 of our qualitative results identified some concerns raised by some physicians which include promoting inter-professional rivalry and encourage unprofessional conduct among lower cadre health staff

This has also be discussed

5 Line 204 -208: This needs to be clarified; the percentages of those who believe trained non-

physician HCWs need or do not need supervision is at variance/ confusing. This has been clarified

6 Line 208 -212: It will be beneficial to also state the percentages here.

5

 This has been clarified.

7 Also, was the level of comprehension/ understanding of the TSTS concept among the

respondents assessed? It will have been useful to compare their level of understanding of the

concept with their perception. Because, if they had a good understanding of the concepts, and

still did not give an enthusiastic endorsement, then there are clear concerns.

 No, it was not assessed. The concept of TSTS was defined and introduced to the respondents before they started filling the question

We agree that their understanding could have been tested and be related with their perception. This has been included as limitation of the study

8 . In addition, it will have been beneficial to narrow the respondents of this study to those who are

involved, in some capacity, in the treatment of patients with hypertension.

 Comments well noted 

All consenting physicians were captured, however, majority(88%) were involved in clinical practice. The perception of physicians who were not directly involved in hypertension management such as those who are involved administration and policy making may also be important for successful implementation of TSTS Strategy

9 6. Line 213 -222: These parameters used for the analysis here are rather generic for the evaluation

of such an important policy.

 The mixed method was aimed at addressing this concern, like said earlier in question1. The usage of bivariate analysis established existence of association, and the qualitative component idea was to attend to the contextual nature of the research idea rather than quantitative assessment only. 

10 What proportion of physicians recruited in this study are involved in clinical work? 1100 (88%) of the physician were in clinical work at the time of the study. It’s presented in line 508 -table .

11 It would have been more informative if the inclusion criteria for this study had been better

defined at the beginning. Now included. 

Only licensed and practicing physicians who were either previously or currently involved clinical work were included in the study. 

12 The discussion highlights the findings of positive perception among the respondents, without

addressing the reasons for the significant proportion of respondents with negative impression.

In order to get the best out of this policy, the concerns of those with negative perception will need

to be highlighted and addressed.

 This has been addressed in Page 10, Line 335 to 343

13 . Also, it is worthy to note that the loss of HCWs in the country is not just among physicians, but

other professionals cadres as well. Yes, we agree that migration of HCWs out of the country affects non-physicians health workers. However, our study is focused on the Physicians because they are the ones whose role in hypertension management will be shifted and/or shared

14 What cadre are the authors proposal in this roll-out? The plan is to assess the perception of non-physician health workers about TSTS. The findings will be useful to make useful recommendations that will facilitate the successful implementation of this policy

---

## [Decision Letter · Decision Letter 2]

1 Sep 2023

Physicians’ Perception of Task Sharing with Non-Physician Health Care Workers in the Management of Uncomplicated Hypertension in Nigeria: a mixed method study

PONE-D-22-34586R2

Dear Author/s,

We’re pleased to inform you that your manuscript has been judged scientifically suitable for publication and will be formally accepted for publication once it meets all outstanding technical requirements.

Kind regards,

Philipos Petros Gile, MA

Academic Editor

PLOS ONE

Additional Editor Comments (optional):

Reviewers' comments:

Reviewer's Responses to Questions

**Comments to the Author**

1. If the authors have adequately addressed your comments raised in a previous round of review and you feel that this manuscript is now acceptable for publication, you may indicate that here to bypass the “Comments to the Author” section, enter your conflict of interest statement in the “Confidential to Editor” section, and submit your "Accept" recommendation.

Reviewer #1: All comments have been addressed

Reviewer #2: All comments have been addressed

Reviewer #3: All comments have been addressed

2. Is the manuscript technically sound, and do the data support the conclusions?

Reviewer #1: Yes

Reviewer #2: Yes

Reviewer #3: Yes

3. Has the statistical analysis been performed appropriately and rigorously? 

Reviewer #1: Yes

Reviewer #2: N/A

Reviewer #3: Yes

4. Have the authors made all data underlying the findings in their manuscript fully available?

Reviewer #1: Yes

Reviewer #2: No

Reviewer #3: Yes

5. Is the manuscript presented in an intelligible fashion and written in standard English?

Reviewer #1: Yes

Reviewer #2: Yes

Reviewer #3: Yes

6. Review Comments to the Author

Reviewer #1: From my side there is no additional comments or questions. I think the authors already addressed the pending issues.

Reviewer #2: (No Response)

Reviewer #3: This manuscript addresses a very important aspect of clinical practice. The authors have made satisfactory efforts to address the concerns that were raised in the initial review.

7. PLOS authors have the option to publish the peer review history of their article (what does this mean?). If published, this will include your full peer review and any attached files.

Reviewer #1: No

Reviewer #2: No

Reviewer #3: **Yes: **Iorhen Akase

---

## [Editor Report · Acceptance letter]

18 Sep 2023

PONE-D-22-34586R2 

Physicians’ Perception of Task Sharing with Non-Physician Health Care Workers in the Management of Uncomplicated Hypertension in Nigeria: a mixed method study 

Dear Dr. Adejumo:

I'm pleased to inform you that your manuscript has been deemed suitable for publication in PLOS ONE. Congratulations! Your manuscript is now with our production department. 

Kind regards, 

on behalf of

Dr. Philipos Petros Gile 

Academic Editor

PLOS ONE